# Natural Products from Mangroves: An Overview of the Anticancer Potential of *Avicennia marina*

**DOI:** 10.3390/pharmaceutics14122793

**Published:** 2022-12-14

**Authors:** Federico Cerri, Marco Giustra, Yaprak Anadol, Giulia Tomaino, Paolo Galli, Massimo Labra, Luca Campone, Miriam Colombo

**Affiliations:** 1Department of Earth and Environmental Sciences, University of Milano Bicocca, 20126 Milan, Italy; 2Department of Biotechnology and Bioscience, University of Milano Bicocca, 20126 Milan, Italy; 3Dubai Business School, University of Dubai, Dubai 14143, United Arab Emirates; 4Marine Research and High Education (MaRHE) Center, Magoodhoo Island, Faafu Atoll 12030, Maldives

**Keywords:** *Avicennia marina*, cytotoxicity, anticancer activity, natural products, bioactive natural compounds, bioprospecting

## Abstract

Exploring the potential of natural extracts for pharmaceutical applications in the treatment of different diseases is an emerging field of medical research, owing to the tremendous advantages that they can offer. These include compound sustainability due to the natural origin and virtually unlimited availability. In addition, they contribute to promoting the countries in which they are extracted and manufactured. For this reason, wild active compounds derived from plants are attracting increasing interest due to their beneficial properties. Among them, *Avicennia marina* has been recently recognized as a potential source of natural substances with therapeutic activities for anti-cancer treatment. *A. marina* beneficially supplies different chemical compounds, including cyclic triterpenoids, flavonoids, iridoids, naphtaquinones, polyphenols, polysaccharides, and steroids, most of them exhibiting potent antitumor activity. The in vivo and in vitro studies on different models of solid tumors demonstrated its dose-dependent activity. Moreover, the possibility to formulate the *A. marina* extracted molecules in nanoparticles allowed researchers to ameliorate the therapeutic outcome of treatments exploiting improved selectivity toward cancer cells, thus reducing the side effects due to nonspecific spread.

## 1. Introduction

Mangrove plants are distributed in 123 tropical and subtropical countries of the world and include 74 species of true mangroves [1]. *Avicennia* is one of the true mangroves from the *Acanthaceae* family [2]. Globally, there are eight species of *Avicennia*, which is the only mangrove genus found worldwide. They are densely distributed mangrove species, found in both coastal rivers and seabeds in tropical and temperate regions [3].

*Avicennia marina (A. marina)* is a mangrove species belonging to the Verbenaceae family, which is widely distributed in the tropical and subtropical zones of the world. They can grow in difficult climatic conditions, such as very windy areas with high temperatures, high salinity, and anaerobic soils. *A. marina,* also known as the “grey mangrove”, grows as shrubs or trees that can reach heights from 3 to 14 m. From a botanical point of view, they have light grey bark formed by thin and delicate scales, while the leaves are green and about 6–7 cm long, with small hairs on the surface below. In the spring, *A. marina* produces small clusters of three to five yellow flowers with a diameter of about one centimeter, and a fruit that contains a large and fleshy seed [4,5].

The plant has a wide geographical distribution. It is distributed along the tropical coasts of East Africa, South and Southeast Asia, Australia, the North Island of New Zealand, and islands of the Pacific Ocean, including Fiji. It is also found in temperate regions, including Southwestern Asia, along the coastline of the Arabian Gulf as well as the eastern and western coasts of the Red Sea [6]. *A. marina* is the most dominant mangrove species, and the dominant coastal vegetation growing along the Arabian Gulf coasts of the United Arab Emirates (UAE), Saudi Arabia, Bahrain, Qatar, and Iran [7,8].

Mangroves are one the most economically and ecologically important ecosystems in the world because of the range of ecosystem services they provide, including enhancement of fisheries, carbon storage and sequestration, and coastal protection against the diverse impact of natural disasters. Moreover, they protect the marine ecosystem and improve the quality of coastal and nearshore waters through nutrient cycling, as well as by neutralizing aquatic and terrestrial pollutants. They also act as a support for marine life (fish, shrimps, crabs, etc.) and wildlife (mammals, reptiles, birds, etc.) species by providing them with a habitat and food [7,9,10]. *A. marina* is a mangrove species that is highly resistant to extreme environmental conditions, such as high salinity, high temperature, strong winds, and anaerobic soil [6,11]. The grey mangrove can experience stunted growth in water conditions that are too saline, but they thrive and reach their full height in waters where both salt- and freshwater are present. The species can tolerate high salinity by excreting salts through its leaves [12,13]. These extraordinary capacities for resistance prompted researchers to investigate the potential of this plant for pharmaceutical uses and the development of new drugs. In particular, the aim of this review is to highlight the anticancer potential of the extracts and bioactive compounds of this plant.

## 2. Components and Biological Activities of *Avicennia marina*

As reported in the literature, chemical investigations of the plant revealed the presence of aliphatic alcohols, amino acids, carbohydrates, alkaloids, carotenoids, fatty acids, hydrocarbons, iridoid glucosides, abietane diterpenoid glycosides, carboxylic acids, steroids, tannins, triterpenes, naphthalene derivatives, flavones, flavonoids, phorbol esters, phenolic and related compounds, pheromones, inorganic salts, minerals, phytoalexins, and vitamins (Figure 1) [14,15,16,17].

Ibrahim et al. [14] analyzed extracts of *A. marina* leaves by GC–MS, and the main constituents were identified and quantified (Table 1).

### 2.1. Naphthalene Derivatives

For the first time, Han et al. [18] reported the occurrence of seven unusual naphthoquinone derivatives isolated from the twigs of *A. marina* collected in Xiamen, called avicennone A–G (**1–7**), together with the known compounds, namely stenocarproquinone B (**8**), avicequinone A (**9**), avicenol A (**11**), and avicenol C (**12**). Compounds **8–10,** containing 4,9-dione group as the main substituent and a mixture of compounds **4** and **5**, showed high antiproliferative and moderate cytotoxic effects, as well as antibacterial activities. The structures of naphthalene derivatives **1–12**, derived from *A. marina*, are displayed in Figure 2.

### 2.2. Flavones

For the first time, Sharaf et al. [19] isolated two flavones, luteolin 7-*O*-methylether 3′-O-β-D-glucoside (**13**) and its galactoside analogue (**14**), from aerial parts of A. *marina,* collected in a region in the south of Hurghada. They found that they exhibited moderate cytotoxicity against the BT-20 human carcinoma cell line (ED50 = 18 µg/mL). Jia et al. [20] identified a new flavone from the leaves, 5-hydroxy-4′,7-dimethoxyflavone (**15**), along with the common flavones quercetin (**16**) and kaemferol (**17**). Feng et al., from aerial parts of *A. marina* harvested in Hainan island, isolated four hydroxylated flavones, including 4′,5-dihydroxy-3′,7-dimethoxyflavone (**18**), 4′,5-dihydroxy-3′,5′,7-trimethoxyflavone (**19**), 4′,5,7- trihydroxyflavone (**20**), and 3′,4′,5-trihydroxy-7-methoxyflavone (**21**) [21]. Their antioxidant activity was evaluated using the DPPH radical-scavenging assay. In particular, compounds 18 and 19 showed low activity levels, while compounds 20 and 21 exhibited moderate levels, with IC50 of 52.0 and 37.0 μg/mL, respectively. Another methoxylated flavone, 5,7-dihydroxy-3′,4′,5′-trimethoxyflavone (**22**), was also isolated by the same researchers in 2007 [22]. The chemical structures of flavones **14–22**, derived from *A. marina,* are shown in Figure 3.

### 2.3. Iridoid Glucosides

In 1985, through the extraction of wild growing plants in Ceylon, Konig et al. [23] obtained seven iridoids from methylated extract of the leaves of *A. marina*. The isolated iridoid compounds were geniposidic acid (**23**), 2′-cinnamoyl-mussaenosidic acid (**24**), geniposide (**25**), mussaenoside (**26**), 2′-cinnamoyl- mussaenoside (**27**), 10-O-(5-phenyl—2,4-pentadienoyl)-geniposide (**28**), and 7-O-(5-phenyl-2,4-pentadienoyl)-8-epiloganin (**29**). This study also shows that iridoids occur as free acids in the plant. Shaker et al. [24] described the isolation of three new geniposidic acid esters, namely 10-O-[(E)-cinnamoyl]-geniposidic acid (**30**), 10-O-[(E)-p-coumaroyl]-geniposidic acid (**31**), and 10-O-[(E)-caffeoyl]-geniposidic acid (**32**), from butanol extract of *A. marina,* which was collected in Hurghanda, Egypt. In order to obtain the pure iridoid compounds, the authors used a combination of three purification steps: silica gel column, reverse phase C18, and Sephadex LH-20, respectively. Feng et al. [25] reported, for the first time, the purification of two new iridoid glucosides, namely 2′-O-[5-phenylpenta-(2E,4E)-dienoyl] mussaenosidic acid (**33**) and 2′-O-(4-methoxycinnamoyl) mussaenosidic acid (**34**), and one known iridoid glucoside, 2′-O-coumaroylmussaenosidic acid (**35**). The compounds were isolated from the dried aerial parts of the mangrove plant *A. marina* and extracted with MeOH/CHCl_3_ 1:1. After the purification, which was performed by a combination of different chromatography techniques, their structures were elucidated using both NMR spectroscopy and low/high resolution mass spectrometry. The compounds **33–35** showed only weak radical scavenging activity, which was evaluated using a DPPH assay.

Through a chemical analysis of the mangrove leaves (*A. marina*) harvested in the coast of Xiamen region, Sun et al. [15] described, for the first time, the isolation and characterization of five iridoid glucosides. These were called marinoids A-E (**36–40**), along with the known iridoids 2′-cinnamoyl-mussaenosidic acid (**24**), 2′-O-(4″-methoxycinnamoyl) mussaenosidic acid (**34**), and 2′-O-(4″-hydroxycinnamoyl) mussaenosidic acid (**26**). The structures of iridoid glucosides **23–40**, derived from *A. marina*, are shown in Figure 4.

### 2.4. Terpenoids

The presence of triterpenoids in *A. marina* bark (betulic acid (**41**) 0.3%, taraxerol (**42**) 0.06%, and taraxerone (**43**) 0.05%), and traces of hydrocarbons was reported by Bell et al. [26]. Jia et al. [20] first isolated lupeol (**44**), betulin (**45**), β-sitosterol (**46**), and ergost-6,22-diene-5,8-epidioxy-3-ol (**47**) from the leaves of *A. marina,* which were collected in Beihai (China). Two new abietane diterpenoids, a pair of inseparable epimers (6Hα-11,12,16-trihydroxy-6,7-secoabieta-8,11,13-triene-6,7-dial 11,6-hemiacetal (**48**) and 6,11,12,16-tetrahydroxy-5,8,11,13-abitetetraen-7-one (**49**)), were found in *A. marina* twigs in Xiamen (China) by Han et al. [27]. These compounds showed moderate cytotoxic and antimicrobial activities. The structures of terpenoids **41–49**, derived from *A. marina*, are shown in Figure 5.

Some of the reported traditional medicinal uses of *A. marina* extract include the treatment of skin disorders, colds, rheumatism, smallpox, ulcers, larynx, and dysentery [16,19,28,29]. Moreover, biological activities of the different extracts or isolated compounds from *A. marina* have been reported, including antimalarial, antibacterial, analgesic, antioxidant, antifouling, and anticancer effects [16,30]. The presence of flavonoids, tannins, and phenols provides antidiabetic properties to *A. marina*, since they are able to induce the insulin secretion mediated by pancreas, resulting in an anti-hyperglycemic action [31]. In addition, the stigmasterol-3-O-β-D galactopyranoside was demonstrated to show antiglycation effect [32]. The furano-naphthoquinone isolated from the heartwood, Avicequinone C, was successfully employed in the treatment of androgenic alopecia thanks to its 5α-R1 inhibitory activity [33]. *A. marina* derivatives are also indicated in the treatment of inflammatory disorders and rheumatoid arthritis due to their regulatory properties. In arthritic rats, they could inhibit the complete Freund’s adjuvant (CFA)-induced skin lesions and articular deformities [34].

*A. marina* has frequently been investigated for anticancer therapy treatments. With a very high incidence, which increases year by year, cancer is the second leading cause of death after stroke, with an estimated 19.3 million new cases and nearly 10 million cancer deaths worldwide in 2020 [35]. The phytochemical research of bioactive natural products has attracted a significant amount of interest. [36] Nearly 60% of the drugs approved for cancer therapy are natural products derived mainly from plants, such as vincristine (VCR), etoposide irinotecan, taxanes, and camptothecins, or from microbes, such as actinomycin D, mitomycin C, bleomycin, doxorubicin, and l-asparaginase [37].

Starting from this basis, in this paper, we focused our attention on the potential anticancer benefits of *A. marina*, reporting all studies that have demonstrated anticancer activity of plant extracts or compounds isolated from them.

## 3. Anticancer Activity of *Avicennia marina*

Two new flavonoids, luteolin 7-O-methylether 3′-O-β-d-glucoside (**13**) and its galactoside analogue (**14**), provided from the methanol extract of aerial parts of *A. marina,* proved to be cytotoxic against cell line BT-20 of human breast cancer, showing ED50 values of 16 and 18 μg/mL, respectively [19].

From the twigs of *A. marina,* seven new naphthoquinone derivatives were isolated: avicennones A−G (**1−7**) together with five known natural products (**8–12**). Tests were undertaken using Avicennone A (**1**), stenocarpoquinone B (**9**), avicequinone C (**10**), avicenol A (**11**), avicenol C (**12**), and a mixture of Avicennone D (**4**) and E (**5**). These were used against L-929 mouse fibroblasts and K562 human chronic myeloid leukemia cells to demonstrate an anti-proliferative effect, as well as cytotoxic activity against the HeLa human cervix carcinoma cell line. Antiproliferative effects were shown against L-929 (GI50 values of 1.2, 0.8, and 4.4 μg/mL, respectively) and K562 (GI50 values of 0.2, 1.1, and 7.5 μg/mL, respectively) by stenocarpoquinone B (**9**), avicequinone C (**10**), and the mixture of avicennone D (**4**) and F (**5**); however, these appeared at lower values compared to the standard paclitaxel (GI50 values of 0.1 and 0.01 μg/mL, respectively). These compounds also proved to be cytotoxic toward the HeLa cell line, with CC50 values of 4.3, 3.2, and 13.1 μg/mL, respectively, which were higher compared with the positive control (CC50 value of 0.01 μg/mL). An interesting point is that all of the active compounds share the *p*-dione of the naphthoquinone core as a structural element. Referring to the other compounds, instead, they have shown little activity against the aforementioned cancer cell lines [18].

Methanolic extract of *A. marina* leaves exhibited cytotoxic activity on human breast MDA-MB 231 cancer cells, with an IC50 value of 480 µg/mL, while it had no significant effect against normal cell line L929. In addition, the extract induced apoptosis in a dose-dependent manner, and also showed a time-dependent growth inhibition effect of 40%, 44%, and 59% after 24, 48, and 72 h of treatment, respectively [38].

Ethanol extract from *A. marina* leaves was found to be cytotoxic against human promyelocytic leukaemia HL-60 cells, with IC50 values of 600, 400, and 280 µg/mL after 24, 48, and 72 h, respectively, in a concentration- and time-dependent manner. Treated cells, compared to the control cells, appeared smaller, less refracted, membrane blebbing, and with more granular material. Moreover, flow cytometric analysis confirmed that apoptosis was the mechanism of cell death induced by the extract, and revealed that a concentration of 600 µg/mL induced 62% apoptosis at 24 h [28].

Methanolic and aqueous extracts of *A. marina* leaves were found to be cytotoxic against human promyelocytic leukemia HL-60 cells (IC50 values of 277.129 and 291.773 µg/mL, respectively) and the human non-small cell lung cancer NCI-H23 cell line (IC50 values of 221.173 and 237.179 µg/mL, respectively) with an effect comparable to that of doxorubicin [39].

The methanolic extract of *A. marina* stem bark exhibited cytotoxicity against HL-60 and NCI-H23 (IC50 values of 297.934 and 210.987 µg/mL, respectively) as well as the aqueous extract was found to be cytotoxic on HL-60 and NCI-H23 (IC50 values of 281.175 and 220.127 µg/mL, respectively). The extracts displayed comparable cytotoxic IC50 values with the standard doxorubicin and showed negligible toxicity against the human embryonic kidney (HEK-293T) normal cell line [40].

Ethyl acetate extract of *A. marina* leaves and stems displayed, after 48 h, 65% and 75% growth inhibition of the breast adenocarcinoma cell line MCF-7, at, 100 µg/mL and 200 µg/mL, respectively. Further, 100 µg/mL of the extract showed 10% apoptosis at 24 h, while no increasing value in apoptosis was found at 48 h or at 72 h. Nevertheless, increasing the extract concentration to 200 µg/mL displayed 25% apoptosis at 24 h, with an increase of 55% and 75% at 48 h and 72 h, respectively [16]. Due to the lack of literature regarding the molecular mechanisms of cell death induced by *A. marina* extracts, Esau et al. decided to focus on the intracellular pathways involved in the apoptotic effect of ethyl acetate extract against the MCF-7 cell line. Their study proved that the 200 µg/mL of extract concentration unleashed ROS-mediated autophagy, as well as caspase-independent apoptosis.

Further studies by Huang et al. [30] focused on the potential association between the phenol and flavonoid contents of water, ethanol, methanol, and ethyl acetate extracts of *A. marina* leaves with their anticancer activities. In vitro experiments were performed on three human breast cancer cell lines (AU565, MDA-MB-231, and BT483), two human liver cancer cell lines (HepG2 and Huh7), and one normal cell line (NIH3T3). The outcomes revealed that the ethyl acetate extract of *A. marina* was the one carrying the highest concentrations of flavonoids and phenolic compounds, and proved, at the same time, that its anticancer activities were the most effective ones. Furthermore, ethyl acetate extract was found to be unable to inhibit the proliferation of NIH 3T3 cells at 40–80-μg/mL. Therefore, subsequent analyses were performed at a concentration range of 40–80 μg/mL following treatments with ethyl acetate extract. In addition, the colony formation in soft agar of AU565, BT483, HepG2, and Huh7 cancer cell lines was reduced after 14–21 days of treatment with the extract. Furthermore, F2-5, F3-2-9, and F3-2-10 ethyl acetate fractions were all obtained by performing column chromatography, and showed higher cytotoxic activity compared to other fractions against AU565, BT483, HepG2, and Huh7 cell lines, displaying IC50 values of 0.75, 0.85, 0.79, and 15.6 μg/mL, respectively. The ^1^H-NMR and ^13^C-NMR profiles demonstrated that the F3-2-10 fraction contained avicennones D (**4**) and E (**5**). The suppression of MDA-MB231 tumor growth in nude mouse xenografts provided by ethyl acetate extract of *A. marina* leaves elicited the suggestion that this extract may be useful in the treatment of breast cancer.

The polyisoprenoids extract from *A. marina* leaves displayed weak cytotoxicity (IC50 value of 154.987 μg/mL) against WiDr colon cancer cells when compared with the standard doxorubicin (IC50 value of 5.445 μg/mL) [41]. In addition, it was found that the mechanism underlying the cytotoxic effect of the extract was due to cell cycle inhibition and induction of apoptosis.

Yang et al. [29] isolated one new triterpenoid saponin, along with 29 known compounds, from the ethanol extract of *A. marina* fruits. The new 6′-O-(n-butanol) ilekudinoside B ester (**50**) (Figure 6) was found to be cytotoxic against two human glioma stem cell lines, GSC-3# and GSC-18#, with IC50 values of 12.21 and 5.53 μg/mL, respectively.

Qurrohman et al. [42] extracted polyisoprenoids from n-hexane extract of *A. marina* leaves. Polyisoprenoids exhibited cytotoxic activity against WiDr cells, with an IC50 value of 295.25 μg/mL, while the standard 5-FU had an IC50 value of 17.43 μg/mL. Cell cycle analysis revealed that the cell cicle inhibition of polyisoprenoids occurred in the G0-G1 phase. Furthermore, RT-PCR revealed that polyisoprenoids downregulated the P13k, Akt1, mTOR, and EGFR gene expression; however, they upregulated P53 gene expression.

The hexane extract of *A. marina* leaves showed cytotoxic activity against human colon HCT-116, human liver HepG2, and human breast MCF-7 cancer cell lines, with IC50 values of 23.7 ± 0.7, 44.9 ± 0.93, and 79.55 ± 0.57 μg/mL, respectively, lower than those of the standard doxorubicin (IC50 values of 0.45 ± 0.052, 0.42 ± 0.10, and 0.6 ± 0.022 μg/mL, respectively). The study also revealed that hexane extract had a weak ability to induce apoptosis, although the cells showed membrane blebbings in addition to apoptotic bodies. Furthermore, it displayed inhibition of the cell cycle in the G0/G1 phase for HCT-116 cancer cells, and in the S phase for HepG2 and MCF-7 cell lines [43].

The different acetone extracts of *A. marina* leaves were prepared in a concentration range of 40–160 mg/mL, and the maximum cytotoxic activity against the liver HepG2 cancer cell line was observed at 120 mg/mL [44].

Eldohaji et al. [17] isolated lupeol (**44**), a pentacyclic triterpenoid, from hexane extract of *A. marina* stems, clarifying its mechanism of anticancer action, since the data reported on lupeol were approximate and contradictory. The results indicated that lupeol caused considerable (*p* < 0.001) growth inhibitory activity on breast MCF-7 (45%), resistant MCF-7 (46%), liver Hep3B (72%), and resistant Hep3B (35%) cancer cell lines, with slight toxic effects on normal fibroblast cells (F180). The mechanism of action of this triterpenoid was investigated by detecting its influence on key actors in cancer development and progression: BCL-2 anti-apoptotic and BAX pro-apoptotic proteins. They found that lupeol significantly (*p* < 0.01) downregulated BCL-2 gene expression in parental and resistant Hep3B cells by 33 and 3.5 times, respectively, contributing to the induction of apoptosis in Hep3B cells, while no consequence of BAX was found. Proteins extracted from lupeol-treated Hep3B cells were analyzed by Western blot, which indicated the presence of activated caspase-3 cleaved by lupeol. Furthermore, as an indication of the absence of immune/inflammatory responses, the compound exhibited a negligible effect on the proliferation of monocytes, but caused an increase in the sub-G1 population and a reduction in the apoptosis rates of monocytes at 48 and 72 h.

Ethanolic extracts (400 μg/mL) of the lower half of pneumatophores, leaves, the upper half of pneumatophores, and shoots of *A. marina* induced cell growth of 50% or more as well as inhibition of liver HepG2 cancer cells. Additionally, the leaf extract proved to be the most cytotoxic [10].

Ethyl acetate extract from *A. marina* roots contains a moderate quantity of phenolic compounds (yield, 2.48%) with no saponin detected in the sample, as well as a moderate amount of flavonoids (yield, 1.59%). The extract exhibited a cytotoxic effect on colorectal HT29, cervical HeLa, and breast T47D cancer cell lines, with IC50 values of 12.17, 22.76, and 163.61 μg/mL, while the positive control, cisplatin, displayed IC50 values of 115.91, 1.86, and 31.08, respectively. Moreover, the extract did not provide a cytotoxic effect against the normal cell line, hADSC [45].

Afshar et al. [46] evaluated the anticancer effects of ethanol and ethyl acetate extracts of *A. marina* leaves. Phytochemical analysis revealed high phenolic and flavonoid contents, showing, for ethanol extract, 345 µg/mL gallic acid/0.01 g extract and 47.8 Mm GAE, and for ethyl acetate extract, 147.5 µg/mL gallic acid/0.01 g extract and 38.6 mM GAE. GC-MS analyses identified 60 compounds in the ethanol extract and 56 compounds in the ethyl acetate extract. The ethanol and ethyl acetate extracts exhibited cytotoxic activity on human breast cancer MCF-7 and human cervical HeLa cancer cells, with CC50 values of 70 and 102 µg/mL and 189 and 67 µg/mL, respectively. On the other hand, the extracts showed less activity against human ovarian carcinoma OVCAR3 cells (CC50 of 1087 and 272 µg/mL) and kidney epithelial Vero normal cells (382 and 242 µg/mL). Additionally, ethanol extract induced cell cycle arrest in the MCF-7 cancer cell line, while ethyl acetate extract caused apoptotic mechanisms in the OVCAR3 and HeLa cancer cell lines. Moreover, Western blot analysis demonstrated the increase in pro-apoptotic cell effectors, such as Bax and caspase-1, -3, and -7.

## 4. *Avicennia marina* Formulation to Improve the Anticancer Therapeutic Effect

The poor efficacy of many cancer treatments is often associated with the low targeting ratio of drugs, and to the side effects on healthy tissues, due to the aspecific distribution into other organs and tissues. Therefore, a much attention has been dedicated to the study of strategies to obtain a site-specific accumulation of therapeutic agents to the tumor region, avoiding side effects and toxicity [47].

In this scenario, nanotechnology-based formulation is one of the most promising approaches exploited to overcome the bottlenecks of aspecific biodistribution, side effects, and low tumor accumulation [48,49]. Nanoparticles (NPs) for drug delivery are carriers in the 1–1000 nm range, composed of different materials, including biocompatible and biodegradable natural/synthetic molecules, polymers, lipids, or metals. NPs are designed and developed to be loaded or covalently linked with bioactive molecules, such as proteins, peptides, antibodies, and nucleic acids, with the aim to: (1) overcome the problems associated with molecules’ solubility and in vivo bioavailability; (2) avoid the molecules’ degradation in the bloodstream; (3) improve the molecules’ targeting, internalization, and accumulation in the desired cells and tissues; (4) potentiate the drug’s effect. [50] For these reasons, NPs are associated with plants and natural compounds to obtain a therapeutic, synergistic effect. Some papers have reported on the use of NPs to overcome the problems associated with the low tumor accumulation of *A. marina,* aiming to improve its therapeutic effect.

Biogenic engineered silver nanoparticles (AgNPs) were synthesized from aqueous extract of *A. marina* leaves by Varunkumar et al. [51]. The resulting AgNPs displayed dose-dependent cytotoxic activity in the A549 lung cancer cell line (IC50 = 50 μg/mL), inducing ROS-mediated apoptosis, which was also confirmed by RT-PCR and Western blotting analysis. Both the p53-dependent and -independent caspase intermediated signaling pathways were demonstrated to be involved in the process. Tian et al. [52] confirmed the anticancer property of synthesized AgNPs in A549 lung cancer cells; they observed a dose-dependent effect based on ROS activity, with an inhibition of 54% at the concentration of 50 µg/mL and 94% inhibition at 80 µg/mL.

## 5. A New Field of Study

The studies reported in this review highlight the fact that the investigation of the anticancer potential of *A. marina* is a recent and promising field of research. In fact, most of the studies refer mainly to the last decade, with the number of publications increasing in recent years. On the other hand, the number of compounds isolated from *A. marina* extracts and tested against cancer cell lines is still limited. This encourages further investigation in this context to find new, natural, active molecules extracted from *A. marina* that can be employed as new drugs in cancer treatments.

## Figures and Tables

**Figure 1 pharmaceutics-14-02793-f001:**
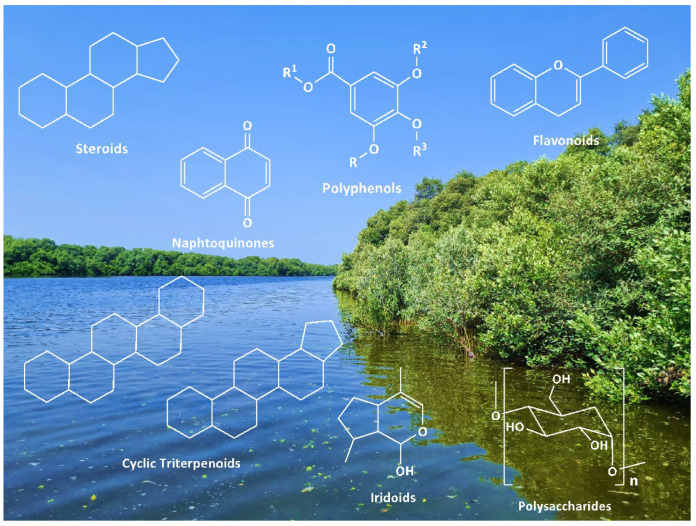
Main classes of compounds extractable from *Avicennia marina*: cyclic triterpenoids, flavonoids, iridoids, naphtaquinones, polyphenols, polysaccharides, and steroids. The picture was taken in the Al Zorah mangrove lagoon (Ajman, UAE).

**Figure 2 pharmaceutics-14-02793-f002:**
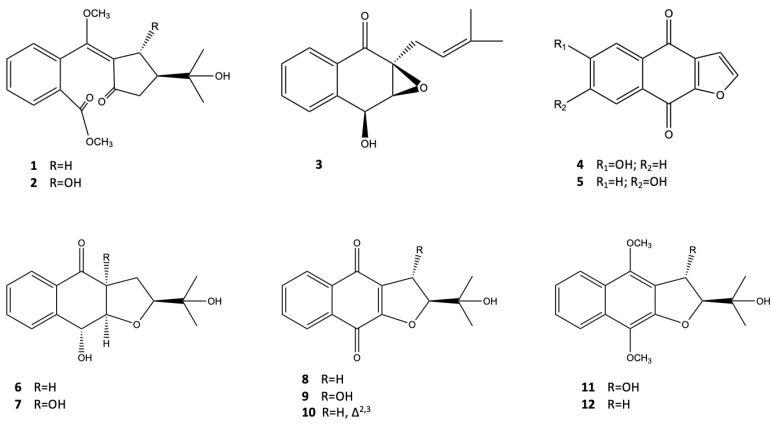
Naphthalene derivatives from *A. marina*.

**Figure 3 pharmaceutics-14-02793-f003:**
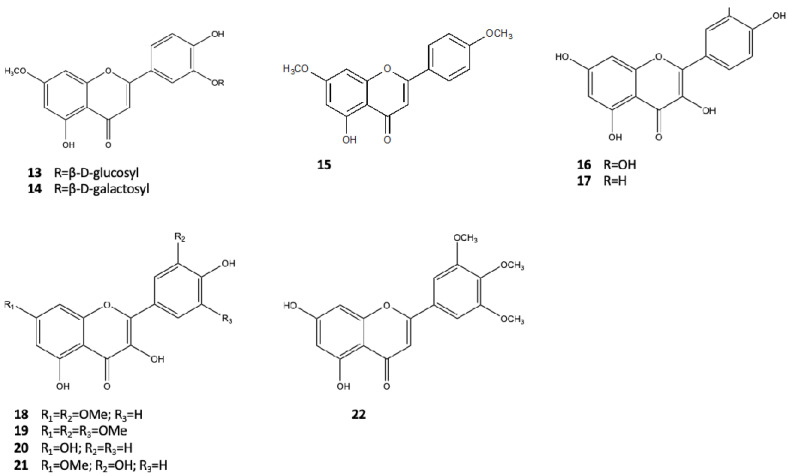
Flavones from *A. marina*.

**Figure 4 pharmaceutics-14-02793-f004:**
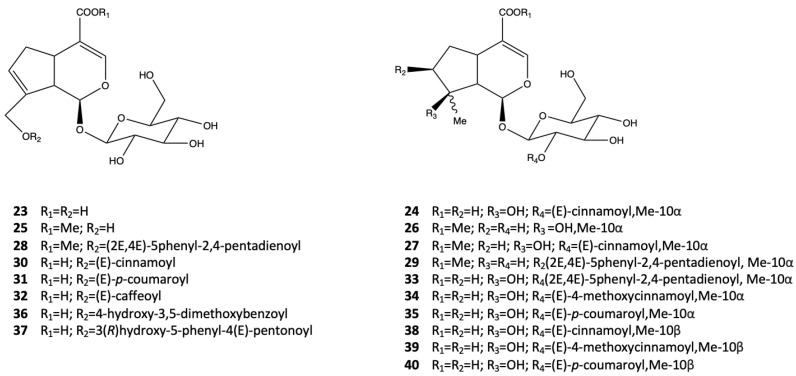
Iridoid glucosides from *A. marina*.

**Figure 5 pharmaceutics-14-02793-f005:**
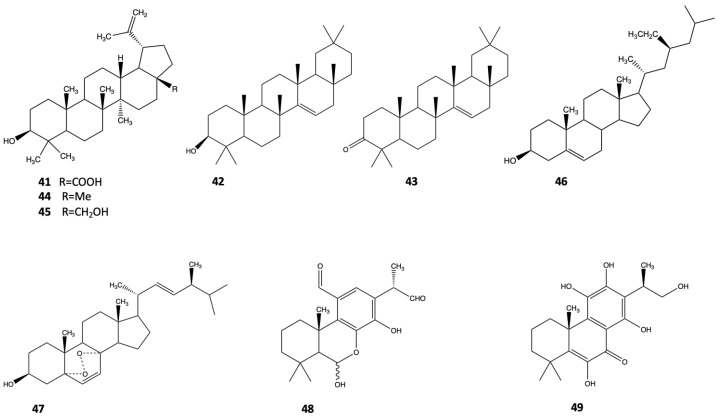
Terpenoids from *A. marina*.

**Figure 6 pharmaceutics-14-02793-f006:**
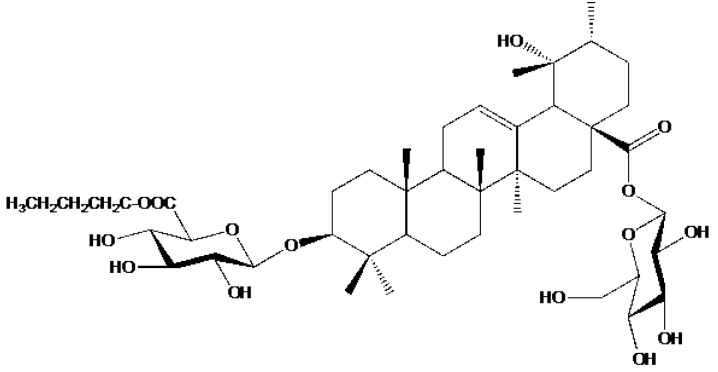
Structure of 6′-O-(n-butanol) ilekudinoside B ester.

**Table 1 pharmaceutics-14-02793-t001:** GC–MS chromatography identified major chemical components of bioactive compounds in the ethyl acetate crude extract of *Avicennia marina* leaves. Adapted from [14].

RT (min)	Name of Compound	Area %	Molecular Formula	Molecular Weight (*m/z*)
4.06	12,15-Octadecadiynoic acid, methyl ester	0.63	C_19_H_30_O_2_	290
7.00	Cyclohexanol,1-methyl-4-(1-methylethenyl)-acetate	3.12	C_12_H_20_O_2_	196
8.87	Undecane	2.36	C_11_H_24_	156
12.82	2-Cyclohexan-1-one, 2-methyl-5-(1-methylethenyl)	1.13	C_10_H_14_O	150
19.76	Bergamotol, Z-α-trans	4.43	C_15_H_24_O	220
21.76	Diethyl phthalate	2.58	C_12_H_14_O_4_	222
23.15	2-Furanmethanol, tetrahydro- α, α,5-trimethyl-5-(4-methyl-3-cyclohexen-1-yl)		C_15_H_26_O_2_	238
23.79	5,8,11,14-Eicosatetraenoic acid, methyl ester, (all-Z)-	2.90	C_21_H_34_O_2_	318
25.18	2H-Pyran-3-ol, tetrahydro-2,2,6-trimethyl-6-(4-methyl-3-cyclohexen-1-yl)-,[3S-[3 α,6 α (R*)]]-	31.13	C_15_H_26_O_2_	238
27.27	E-8-Methyl-9-tetradecen-1-ol acetate	0.91	C_17_H_32_O_2_	268
27.74	1,2-Benzenedicarboxylic acid, bis(2-methylpropyl) ester	5.26	C_16_H_22_O_4_	278
28.01	(E)-Tonghaosu	1.45	C_13_H_12_O_2_	200
28.94	Pentadecanoic acid, 14-methyl-, methyl ester	1.03	C_17_H_34_O_2_	270
29.71	Hexadecanoic acid	2.59	C_16_H_32_O_2_	256
32.15	Ethyl (9z,12z)-9,12-octadecadienoate	1.09	C_20_H_36_O_2_	308
32.28	9-Octadecenoic acid (z)-	2.05	C_18_H_34_O_2_	282
32.51	Phytol	2.33	C_20_H_40_O_2_	296
33.04	12-Methyl-E,E-2,13-octadecadien-1-ol	1.93	C_19_H_36_O	280
39.81	1,2-Benzenedicarboxylic acid	4.36	C_24_H_38_O_4_	390
42.53	Campesterol	1.90	C_28_H_48_O	400
43.24	Stigmasterol	3.89	C_29_H_48_O	412
44.08	α-Sitosterol	11.58	C_29_H_50_O	414

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
