# Peer review of "Natural Products from Mangroves: An Overview of the Anticancer Potential of Avicennia marina"

_pharmaceutics, 2022, doi:10.3390/pharmaceutics14122793_

Round 1
Reviewer 1 Report
Manuscript title: Natural products from Mangroves: an overview of the anticancer potential of Avicennia marina
In the manuscript, the authors made an overall review work on the investigation of the phytochemicals of Avicennia marina in the effects of physiological and pathological studies in vitro and in vivo. In general, the authors have completed a reasonable study with very informative data on the relationships of the phytochemicals and bioactivities. However, the presentation of this study may be strengthened by adding a fingerprint (published or unpublished) of the HPLC analysis on the typical solvent extract of Avicennia marina and relatively marked compound names.
Specific comments:
1. In Figure 6
Please re-confirm that the n-butanol linkage at the left side of the molecular structure.
Author Response
Dear Editor and dear reviewers,thank you for giving us the chance to improve the quality of our manuscript. We have amended the manuscript carefully tanking all the reviewers’ suggestion into account. With the help of their valuable comments, we believe that this revised version of our manuscript has been significantly improved in terms of readability a data presentation.
Best regards,
Miriam Colombo

Reviewer 2 Report
The review concerns the study of anticancer potential of mangroves plant Avicennia marina survived in condition of strong salinity that leads to production of very impressive set of secondary metabolites – natural substances having very interesting therapeutic activities including anti-cancer one. A. marina contains different chemical substances including cyclic triterpenoids, flavonoids, iridoids, naphtaquinones, polyphenols, polysaccharides, and steroids. The most of these metabolites possess potent antitumor activity on different model solid tumors in vitro and in vivo. The creation of nanoparticles with A. marina extracts allows to improve the selectivity toward cancer cells and decrease the side effects. The authors concluded that anticancer potential of A. marina is very promising and may be a serious field of research. Nevertheless, the number of compounds isolated from A. marina extracts and tested against cancer cell lines is not so match and new active natural product should be found from this source.
The article is very well written, all the noted natural products have own unique number using in different places of the text that significantly improve its readability. Only several minor corrections are necessary:
. 1. Page 8, Figure 6. Formula 50. The formula should be corrected at C-6’ position where normal ester of glucuronic acid with N-butanol having terminal methyl group should be presented.
2. Page 9, Line 1 from the top. Insert, please, the number 44 (in bold) in brackets after “lupeol”.
3. All the Latin names of organisms should be presented by italic in the titles of the references.
The review may be published after very minor correction without additional excessive correspondence.
Author Response
Dear Editor and dear reviewers,
thank you for giving us the chance to improve the quality of our manuscript. We have amended the manuscript carefully tanking all the reviewers’ suggestion into account. With the help of their valuable comments, we believe that this revised version of our manuscript has been significantly improved in terms of readability a data presentation.
Best regards,
Miriam Colombo
